# Numerical Study and Structural Optimization of Vehicular Oil Cooler Based on 3D Impermeable Flow Model

**Jiahong Fu** [1,2,*] , **Zhecheng Hu** [2] , **Yu Zhang** [2] and **Guodong Lu** [1,*]

[1] Zhejiang Yinlun Machinery Co., Ltd., Taizhou 317207, China
[2] Department of Mechanical Engineering, Zhejiang University City College, Hangzhou 310015, China; 31903121@stu.zucc.edu.cn (Z.H.); zhangyu_gcxy@zucc.edu.cn (Y.Z.)
* Correspondence: fujh@zucc.edu.cn (J.F.); luguodong@yinlun.cn (G.L.); Tel.: +86-13600542575 (J.F.)

**Abstract:** A non-uniform permeable flow numerical model of vehicular oil cooler was proposed to simulate the thermal performance of oil cooler, due to the complex internal structure of cooler and the anisotropy of coolant flow and heat transfer. By comparing the numerical simulation results with the experimental results, the maximum error of the simulation results under different working conditions is 9.2%, which indicates that the modelling method is reliable and can improve the development efficiency. On this basis, through the three-dimensional numerical simulation to establish and optimize the oil cooler's parameters. The thermal performance under different structural oil cooler were compared using the comprehensive evaluation factor $j/f$. The results and the experimental data show that under the impermeable flow model can obtain good heat transfer efficiency with low flow resistance at the same time. When the cross-sectional area is 3 mm$^2$, length of 90 mm, layer number of 11, the model accuracy was 0.6%, as the optimal structure parameters, the heat transfer increase by 47% and with the total pressure drop increased by only 30%.

**Keywords:** non-permeate flow model; oil cooler; numerical simulation; $j/f$ factor; heat transfer performance

## 1. Introduction

Oil cooler is mainly used in automobile, construction machinery and other vehicle cooling system. The efficient oil cooler can meet the heat transfer requirements of each subsystem of the vehicle, so that each component of the vehicle can have a good thermal state under different operating conditions and environmental conditions. The flow and heat transfer characteristics of oil cooler are important factors affecting heat transfer. With the development of computational fluid dynamics, numerical simulation technology has become a means of researching and developing oil cooler. Due to the complex structure and high precision requirements of the plate-fin, disc and integral oil coolers on the market, a lot of energy and resources are needed for the accurate simulation [1–3], which greatly affects the optimization cycle of its structure parameters.

For instance, when building the overall model of the plate-fin cooler, the scale of the fin structure of the cooler is usually at the millimeter level, but the flow of the heat exchanger may often be at the meter level, resulting in a gap of four orders of magnitude between the length of the flow channel of the heat exchanger and the fin. At this point, to build the entire model of the heat exchanger, the number of grids in unit channel of a heat exchanger reaches 100,000, consequently, the number of grids in the overall heat exchanger even exceeded 10 million. This brings great challenges to the accurate flow and heat transfer simulation of heat exchanger, which not only puts forward high requirements for computing resources, but also has a long calculation cycle.

At present, the research on numerical simulation technology of vehicular heat exchanger is mainly divided into three categories. The first category is the theoretical equivalence of heat exchanger based on empirical correlation [4]. Aydin used approximate

formula to fit the relationship between heat exchanger performance index and structural parameters [5]. Mortean predicted the performance of heat exchanger through the correlation of flow and heat transfer performance, but its simulation accuracy was only about ±20% [6]. Qiu used performance-heat transfer Unit number method ($\varepsilon$-NTU) and Davenport's modified equation to build a flow heat transfer analysis model for heat exchangers, but the accuracy of the model was only between −10% and +24% [7]. Deng fitted the test correlation of outer fins of automotive heat exchangers through experiments to improve the overall performance of heat exchangers [8]. However, the test data came from bench tests, and the prediction of $j$ and $f$ factors of actual heat exchangers needs to be further verified. Therefore, the high precision fitting of empirical correlation depends on a large number of experimental results or the full scale numerical simulation of heat exchangers, and the universality of different heat exchangers is still a problem due to the different fin structure parameters and operating conditions.

The second method is the numerical simulation equivalence of heat exchanger based on the porous media model [9]. The equivalence of heat exchanger in this method is mainly used for the equivalence of flow resistance, but it is still not accurate in the equivalence of heat transfer performance, so the one-dimensional heat transfer model of heat exchanger is mainly established depending on experimental results. This method can not only obtain more accurate three-dimensional flow information of cooling air, but also efficiently calculate the heat transfer performance of heat exchanger. Zhou simulated the resistance characteristics of heat exchanger by using the porous medium model [10]. Du simplified the internal flow channel of plate-fin heat exchanger with staggered teeth by using the porous medium model [11]. Wang and Teodor used the porous medium model to simulate its flow characteristics and the one-dimensional heat transfer matrix of the heat exchanger to simulate its heat transfer characteristics, and the simulation results were in good agreement with the experimental results [12,13]. In addition, Mongibello, Shen, and Yu also adopted this model when conducting the equivalence of heat exchangers [14–16].

The third is multi-scale equivalence of heat exchanger with both micro fin parameters and macro performance parameters [17]. Su analyzed the overall heat transfer performance by means of microscale unit simulation analysis and full-scale model [18]. Huang improved the accuracy of simulation by using multi-scale porous media model [19]. Saravanan and Sethuramalingam established a geometric model through the periodic boundary and calculated the aerodynamic resistance and heat transfer performance of the heat exchanger, which was pointed out that the simulation accuracy of the method for air flow resistance was 8.64%, but did not specify the simulation error of heat transfer performance [20,21]. Starace also constructed gas-liquid side fin element and carried out coupled heat transfer calculation, and established fitting correlation through regression analysis method, which further improved the accuracy of equivalent model [22]. Greiciunas also constructed the porous media unit model, but simplified the source term for heat transfer to improve the calculation efficiency [23]. On this basis, the iteration of import and export boundary conditions is carried out through the periodic boundary, so as to realize the flow and heat transfer simulation of the overall model through the element model. However, this method does not take into account the influence of the import and export header of the heat exchanger on the non-uniform flow distribution, and there is still space for further improvement [24].

Therefore, in this paper, the non-permeable flow model is used to simulate the anisotropic flow inside the oil cooler, and the equivalent simulation method of oil cooler was proposed and verified by experiments, so as to get rid of the dependence on experimental results and significantly improve its calculation efficiency and accuracy. On this basis, the flow and heat transfer performance under different structural parameters are analyzed and optimized.

## 2. Equivalent Theory

### 2.1. Non-Uniform Permeable Flow Model

Flow in porous media is controlled by continuity equation and momentum equation, which together constitute Brinkmann equation [25]:

Brinkman equations:

$$\frac{1}{\varepsilon_p}\rho(\mathbf{u} \cdot \nabla)\mathbf{u}\frac{1}{\varepsilon_p} = \nabla \cdot [-p\mathbf{I} + \boldsymbol{\kappa}] - \left(\mu\kappa^{-1} + \beta\varepsilon_p\rho|\mathbf{u}| + \frac{Q_m}{\varepsilon_p^2}\right)\mathbf{u} + \mathbf{F} \tag{1}$$

Definition of porosity matrix:

$$\boldsymbol{\kappa} = \mu\frac{1}{\varepsilon_p}\left(\nabla\mathbf{u} + (\nabla\mathbf{u})^{\mathrm{T}}\right) - \frac{2}{3}\mu\frac{1}{\varepsilon_p}(\nabla\mathbf{u})\mathbf{I} \tag{2}$$

The definition of parameter $\beta$:

$$\beta = \frac{c_F}{\sqrt{\kappa}} \tag{3}$$

where, $\mu$ is the dynamic viscosity of the fluid, kg/(m·s); $\mathbf{u}$ is the velocity vector, m/s, $\rho$ is the density of the fluid, kg/m$^3$; $p$ is the pressure, $\mathbf{I}$ is identity orthogonal matrix, $\varepsilon_p$ is the porosity, $\boldsymbol{\kappa}$ is the penetration matrix, m$^2$; $Q_m$ is the quality of the source term; $\mathbf{F}$ is volume force, kg/(m$^2$·s$^2$).

### 2.2. Local Thermal Non-Equilibrium Model

In the oil cooler, the heat transfer process of coolant follows the energy conservation equation:

$$\rho c_p\frac{\partial T}{\partial t} + \rho c_p\mathbf{u} \cdot \nabla T + \nabla \cdot (-k\nabla T) = Q \tag{4}$$

The assumption of local heat balance is adopted in the heat transfer equivalence of oil cooler, and the temperature between solid and liquid is considered equal at the interface of solid, namely $T_f = T_s = T$. In steady-state conduction problems, or where the internal volume heating of both materials is the same, the local heat equilibrium hypothesis can be used for most low-response problems, assuming that the solid and fluid temperatures are equal. When the heat transfer of porous media is considered, $\rho c_p$ in the transient term of energy Equation (5) becomes effective volume heat capacity at constant pressure, which is defined as follows:

$$\left(\rho c_p\right)_{\mathrm{eff}} = \theta_s\rho_s c_{p,s} + \varepsilon_p\rho_f c_{f,s} \tag{5}$$

where, $\varepsilon_p$ is the porosity, $\theta_s$ is the volume fraction of solid, and the thermal conductivity term $\nabla \cdot (-k_{\mathrm{eff}}\nabla T)$ is from $\nabla \cdot (-k\nabla T)$, where is the weighted average of the thermal conductivity of solid $k_s$ and liquid $k_f$:

$$k_{\mathrm{eff}} = \theta_s k_s + \varepsilon_p k_f \tag{6}$$

## 3. Numerical Model of Oil Cooler

### 3.1. Heat Exchange Unit Model

Figure 1 shows the fin structure of the oil cooler, from which it can be seen that both the oil side and the water side are staggered tooth fins. Staggered fin has two directions of high resistance flow and low resistance flow at the same time, and the flow direction is random, so it cannot be equivalent to the traditional porous media set only one mainstream direction, and the resistance of the other two directions is set 1000 times of the mainstream direction. In staggered fins, the flow is three-dimensional, and there is no flow in the z direction perpendicular to the wall, but flows in either x or y directions [26].

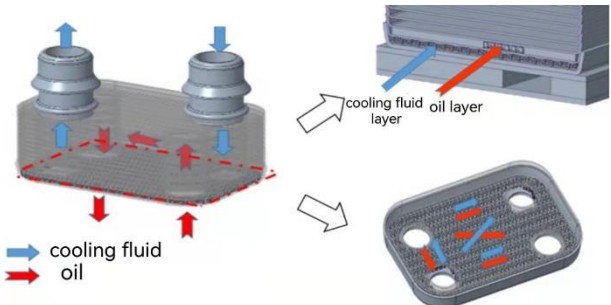

**Figure 1.** Water side fin structure (**left**) and oil side fin structure (**right**).

The three-dimensional flow structure and simulation model are shown in Figure 2. The difference between them mainly lies in the different velocity gradients generated under different pressure gradients, resulting in different velocity components in the x direction and y direction. Therefore, this project proposes to calculate the flow in the direction of high resistance and low resistance simultaneously under the same structure, and consider the change of velocity gradient and viscosity to synthesize the permeability $\kappa$ in two different directions of x direction and y direction, so as to establish the permeability matrix $\kappa = \begin{bmatrix} \kappa_x & 0 & 0 \\ 0 & \kappa_y & 0 \\ 0 & 0 & \kappa_z \end{bmatrix}$ of $x$, $y$ and $z$ anisotropically. This permeability matrix can be used to equivalent flow resistance on both sides of oil and water in the overall model of the oil cooler [27].

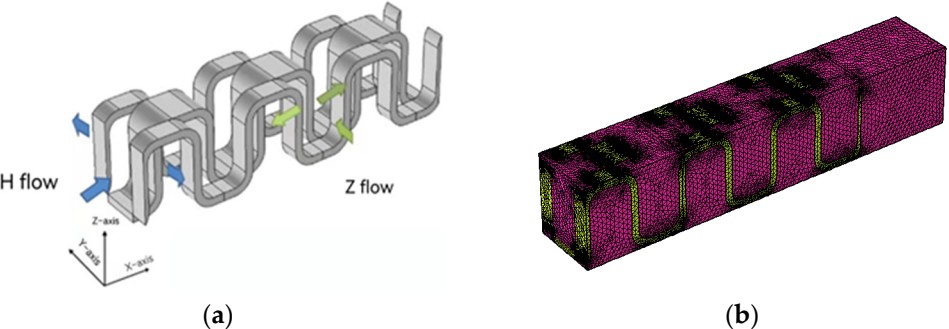

|   (**a**)   |   (**b**)   |

**Figure 2.** Three dimensional geometry structure of staggered fin unit model. (**a**) geometric model; (**b**) simulation model.

### 3.2. Grid Dependence Analysis

Mesh size will have a certain impact on the results, too few grids will lead to poor accuracy of the calculation results, but too dense grids will increase the calculation cost and accuracy is limited, so grid independence verification must be carried out to obtain the comprehensive accuracy and calculation cost of grid number selection As can be seen from Figure 3, with inlet and outlet pressure drop and calculation time as evaluation indexes, after the number of grids reaches 600,000, the calculation time increases significantly with the increase of the number of grids, but the increase of calculation results is not obvious. Based on the calculation results and efficiency, the number of grids selected for H-fin and Z-fin is 600,000.

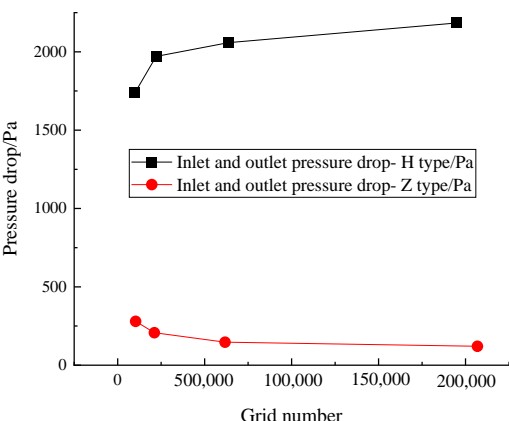

**Figure 3.** Grid independence verification.

### 3.3. Nonlinear Fitting Correlation

In order to obtain the calculation results at different flow rates, the flow rates were set from 0 to 2 m/s at intervals of 0.5 m/s. To simulate the coolant viscosity of different species and at different temperatures, the viscosity was set from 0 to 0.5 Pa·s at 0.05 Pa·s intervals.

Based on the above content, the calculation result is shown in Figure 4 The nonlinear fitting correlation between pressure gradient and viscosity is obtained. On this basis, anisotropic permeability matrix κ can be obtained through calculation of H direction and Z direction respectively, as shown in Equation (7). In macro heat exchanger model, κ and $\beta$ were the key parameters to characterize the flow resistance of the oil cooler.

$$\Delta p = \frac{\mu}{\kappa}u + \beta\varepsilon_p\rho u^2 \tag{7}$$

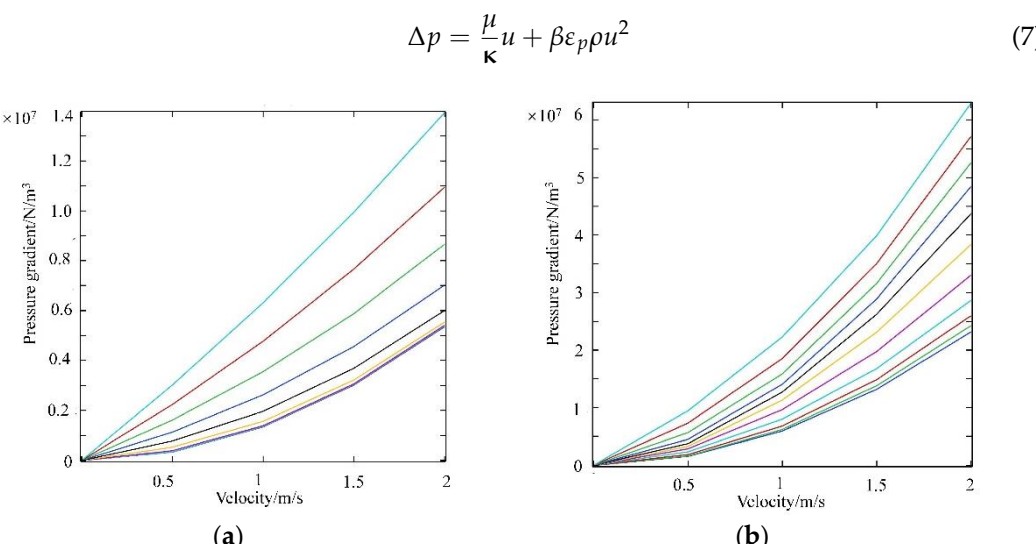

**Figure 4.** The calculation results of H direction and Z direction. (**a**) H direction; (**b**) Z direction.

### 3.4. Establishment of Equivalent Model

In Section 2.1, the connection between the micro fin unit of heat exchanger and the macro performance has been established, so that the micro fin research of heat exchanger and the macro performance research can complement each other. In this section, the equivalent model of oil cooler is designed. Figure 5 reflect the structural size and macroscopic equivalent model of the oil cooler.

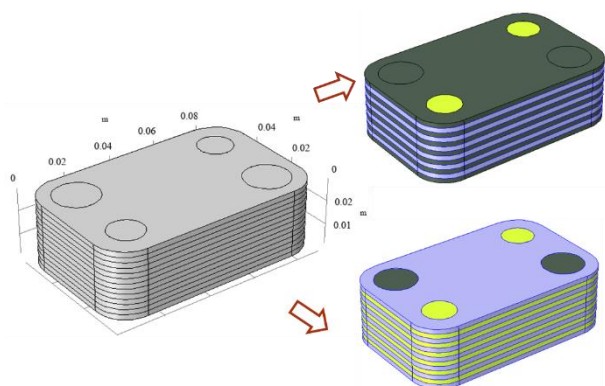

**Figure 5.** Equivalent model of oil cooler.

### 3.5. Boundary Conditions and Thermophysical Parameters

In order to obtain the flow and heat transfer characteristics of the equivalent model of oil cooler under common working conditions, operating parameters such as oil-side inlet temperature, cold-side inlet temperature, oil flow rate and cold-side flow rate are set as shown in Table 1, which are consistent with the experimental verification working conditions.

**Table 1.** Equivalent simulated operating conditions.

| Number | Oil Inlet Temperature (°C) | Cold Test Inlet Temperature (°C) | Oil Flow Rate (kg/min) | Cold Side Flow Rate (kg/min) |
|---|---|---|---|---|
| 1 | 100 | 70 | 12.25 | 12.75 |
| 2 | 100 | 70 | 12.25 | 15.75 |
| 3 | 100 | 70 | 6.75 | 8.75 |
| 4 | 100 | 70 | 6.75 | 12.75 |
| 5 | 100 | 70 | 6.75 | 16.00 |
| 6 | 100 | 70 | 10.00 | 16.00 |
| 7 | 100 | 70 | 10.00 | 13.00 |
| 8 | 100 | 70 | 10.00 | 8.25 |
| 9 | 100 | 70 | 12.75 | 8.25 |
| 10 | 130 | 90 | 12.00 | 15.25 |
| 11 | 130 | 90 | 12.00 | 12.25 |
| 12 | 130 | 90 | 12.00 | 8.25 |
| 13 | 130 | 90 | 9.50 | 8.00 |
| 14 | 130 | 90 | 10.00 | 12.50 |
| 15 | 130 | 90 | 10.00 | 15.50 |
| 16 | 130 | 90 | 6.50 | 15.50 |
| 17 | 130 | 90 | 6.50 | 12.50 |
| 18 | 130 | 90 | 6.50 | 8.00 |

Because the constant pressure heat capacity, density, dynamic viscosity and thermal conductivity involved in the energy equation are all functions of temperature, it is necessary to establish the curves of these thermal physical properties with temperature. Thermal properties of the cold-side medium 50% ethylene glycol +50% pure water are shown in Table 2. The thermal physical parameters of 5W30, a medium heat conduction oil on the hot side, are shown in Table 3.

**Table 2.** Equivalent simulated operating conditions.

| | |
|---|---|
| Density /kg/m³ | $y = 2 \times 10^{-6}x^3 - 0.002x^2 - 0.4554x + 1074.6 \quad R^2 = 0.9999$ |
| Constant pressure heat capacity/J/(kg °C) | $y = 2 \times 10^{-5}x^3 - 0.0051x^2 + 4.207x + 3256.5 \quad R^2 = 0.9989$ |
| Dynamic viscosity/Pa · s | $y = -9 \times 10^{-14}x^5 + 8 \times 10^{-11}x^4 - 2 \times 10^{-8}x^3 + 3 \times 10^{-6}x^2 - 0.0003x + 0.0084$ <br> $R^2 = 0.9989$ |
| Coefficient of thermal conductivity/W/(m · °C) | $y = 7 \times 10^{-7}x^2 - 0.0003x + 0.4423 \quad R^2 = 0.9991$ |

**Table 3.** Equivalent simulated operating conditions.

| | |
|---|---|
| Density /kg/m³ | $y = -8 \times 10^{-5}x^2 - 0.5779x + 898.75 \quad R^2 = 1$ |
| Constant pressure heat capacity/J/(kg °C) | $y = 0.0014x^2 + 4.078x + 1801.4 \quad R^2 = 1$ |
| Dynamic viscosity/Pa · s | $y = -1 \times 10^{-10}x^5 + 6 \times 10^{-8}x^4 - 1 \times 10^{-5}x^3 + 0.0009x^2 - 0.0356x + 0.5894$ <br> $R^2 = 0.9972$ |
| Coefficient of thermal conductivity/W/(m · °C) | $y = 2 \times 10^{-8}x^2 - 1 \times 10^{-4}x + 0.1464 \quad R^2 = 1$ |

The calculation time was 0.2 h on a 128 core workstation with 192 G memory. Solid-Works was used for the geometric model, COMSOL 5.6 was used for the flow field simulation, and pressure-based coupling solver was used for the numerical solution of the governing equation.

## 4. Experimental Verification

### 4.1. Experimental Rig Construction and Error Analysis

The oil cooling test bench is a closed loop system, as shown in Figure 6. It is mainly composed of temperature sensor, pressure sensor, mass flow sensor, electric pump group, electric heater, flow control valve and circulation pipeline. The inlet and outlet pipelines of the cold side and hot side were equipped with 0.2-class Pt100 thermal resistance, 0.5-class pressure sensor and 0.15-class Coriolis mass flow sensor to detect the temperature, pressure and flow of the fluid respectively. The detailed information of the experimental bench can be referring to references [28,29].

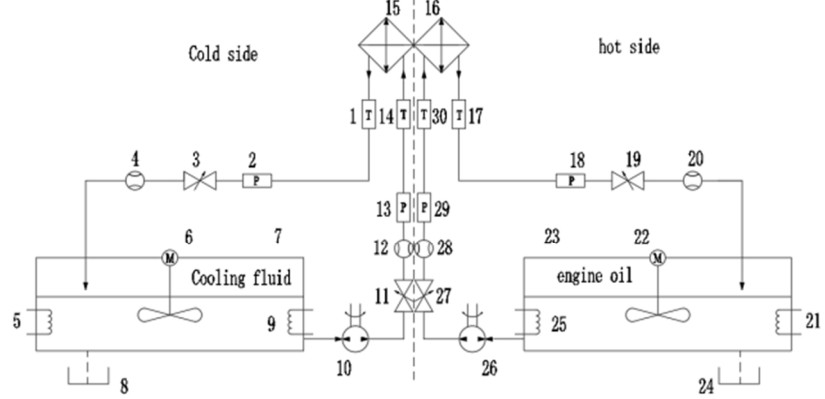

1, 14, 17, 30-Temperature sensor; 2, 13, 18, 29-Pressure transducer; 3, 11, 19, 27-Flow regulating valve; 4, 12, 20, 28-Mass flow sensor; 5, 9, 21, 25-Electric heater; 8, 24-Drainage pipeline、oil pipeline; 6, 22-stirring apparatus; 7, 23-Liquid storage tank; 10, 26-Electric pump group; 15, 16-Cold side and hot side of oil cooler

**Figure 6.** Schematic diagram of the oil cooling test bench.

In this paper, the heat transfer $Q$ needs to be calculated by the mass flow rate and temperature difference of the fluid, and the calculation formula is as follows:

$$Q = c_p m (T_{out} - T_{in}) \tag{8}$$

Therefore, the uncertainty of heat transfer calculation can be expressed by the following formula:

$$\Delta Q = \sqrt{\left(\frac{\partial Q}{\partial m}\delta m\right)^2 + \left(\frac{\partial Q}{\partial \Delta T_1}\delta T_1\right)^2 + \left(\frac{\partial Q}{\partial \Delta T_2}\delta T_2\right)^2}$$
$$= Q\sqrt{(\frac{\delta m}{m})^2 + (\frac{\delta T_1}{\Delta T_1})^2 + (\frac{\delta T_2}{\Delta T_2})^2}$$

(9)

The relative error expression of heat transfer can be obtained:

$$\frac{\delta Q}{Q} = \sqrt{(\frac{\Delta m}{m})^2 + (\frac{\delta T_1}{\Delta T_1})^2 + (\frac{\delta T_2}{\Delta T_2})^2}$$

(10)

The relative error value is 3.2%, which meets the accuracy requirements of the test.

### 4.2. The Results Discussed

The heat balance test of oil cooler was carried out on the experimental platform of heat exchanger flow and heat transfer performance and compared with the numerical simulation results. 5W30 was used as the hot side medium and 50% ethylene glycol +50% pure water was used as the cold side medium. The test conditions were set as Table 1. Figure 7 shows the comparison results of the pressure drop between the oil side and the cold side between test and simulation.

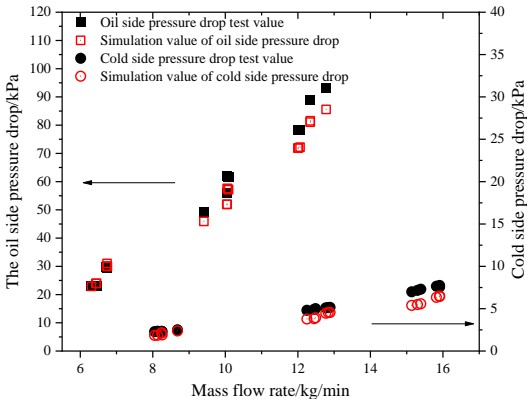

**Figure 7.** The pressure drop of test and simulation varies with the flow rate of oil side and cold side.

As can be seen from Figure 7, when the flow rate is low, the calculation results of the equivalent model can basically correspond to the test results for the flow characteristics in the oil cooler. With the increase of the flow rate, the accuracy of the numerical simulation decreases to some extent, but it is still within the range of credibility.

In order to verify the effectiveness of the whole model for heat transfer process simulation, the heat transfer obtained from experimental conversion and numerical simulation are studied, and the results are shown in Figure 8.

It can be seen from Figure 8 that the heat transfer results obtained by numerical simulation and experiment basically fit, and the errors are all within an acceptable range, with the maximum error of 9.2%. Therefore, it can be concluded that the model can simulate the actual situation of oil cooler. Further analysis shows that when the temperature of the selected working condition is 130 °C, the error of simulation and test of heat transfer increases to a certain extent, which may be caused by the great influence of temperature boundary conditions on the model. In general, the parameter setting in the analysis process is also in line with the actual situation, and the equivalent model has high accuracy in the simulation results of macro performance, which can accurately predict the flow and heat transfer process of the oil cooler.

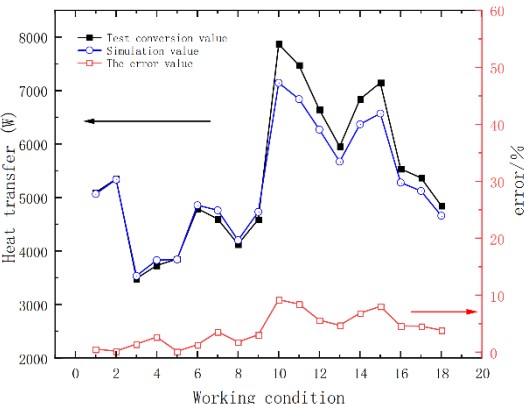

**Figure 8.** Comparison of heat transfer between experiment and simulation.

## 5. Result and Discussion

For oil cooler, flow resistance and heat transfer are two important performance evaluation criteria. For plate-fin oil cooler, the flow performance is mainly divided into two aspects: first, in the same flow channel, uniform fluid velocity is conducive to fin heat transfer; The second is the uniformity of fluid velocity in different layers. In the calculation process, in order to simplify the calculation, the fluid flow velocity of different layers is usually approximately equal, but in the actual working condition, the flow velocity between different layers is different, so in this simulation, the flow performance of these two aspects should be studied. In this analysis, a friction factor *f* is introduced, as shown in Formula (11):

$$f = \frac{2\Delta p A}{\rho v_m^2 L} \tag{11}$$

where, $\Delta p$ is the pressure difference on both sides of the flow passage, *A* is the cross-sectional area of the flow passage, $\rho$ is the density of the liquid, $v_m$ is the average velocity of the fluid in the flow passage, and *L* is the length of the flow passage. It can be seen that several factors affect flow performance: the cross-sectional area and the length of the flow passage. Since it is not easy to directly observe the oil cooler, the dispersion trend needs to be quantified. Therefore, variance, a statistic that can represent the dispersion trend of data, is selected to represent the flow uniformity. The heat transfer performance was evaluated according to the definition of heat transfer, and then the heat transfer performance under the same flow condition was evaluated according to the dimensionless heat transfer *j* factor.

And according to the definition of heat transfer (12):

$$Q = K \cdot A \cdot \Delta T \tag{12}$$

where, *Q* is the total heat transfer of the heat exchanger, *K* is the heat transfer coefficient of the fluid, *A* is the heat transfer area, and $\Delta T$ is the temperature difference between the inlet and outlet of the hot side of the oil cooler.

Heat exchange *j* factor is used to evaluate the heat exchange performance of oil cooler [29], as shown in Formula (13):

$$j = \frac{Nu}{\text{Re} \cdot \text{Pr}^{\frac{1}{3}}} \tag{13}$$

where *Nu* is the Nusselt number, Re is the Reynolds number, Pr is Prandtl number.

The ratio of *j* and *f* factors can be used to comprehensively evaluate the comprehensive effect between heat transfer performance and flow performance. When the oil cooler is started, the index *j*/*f* can be used to quantitatively evaluate the comprehensive performance at the same flow rate.

### 5.1. Flow Heat Transfer Performance Analysis

For plate-fin oil cooler, the flow performance is mainly divided into two aspects: first, in a flow channel, the distribution of fluid velocity in different areas, in the same flow channel, uniform fluid velocity is conducive to fin heat transfer; Secondly, the uniformity of fluid velocity in different layers; In studying the influence of structural parameters on heat transfer performance, the structural parameters in Table 4 will be used for parameter setting of oil cooler. According to the *j* factor, the heat transfers under different runner lengths of 6 mm and 60 mm in height and width can be obtained. The change of heat transfer with runner lengths is shown in Figure 9a. It can be obviously seen that the total heat rejection of oil cooler increases with the increase of runner lengths. It is easy to understand that with the increase of the length of the flow passage, the contact area of the flow passage is also increasing, so as to the total contact area.

**Table 4.** Overall model parameter.

| The Length of the Channel | Channel Width | Oil Domain Channel Height | Water Channel Height | Number of Layer |
|---|---|---|---|---|
| 90 mm | 60 mm | 6 mm | 6 mm | 6 |

**Figure 9.** Heat exchange at different runner heights and lengths:(**a**) length; (**b**) height.

The influence of height changes on heat transfer of oil cooler is shown in Figure 9b. It can be seen that under the premise of constant channel length, the heat transfer decreases with the increase of channel height. This is because when the flow rate is constant, the flow velocity decreases with the increase of the height, and the disturbance of the corresponding flow velocity on the wall also decreases. In addition to these influencing factors, during the operation of plate-fin oil cooler, the shell side flow will pulsate, and with the increase of pulsation frequency and relative amplitude, the power consumption will increase accordingly. Therefore, with the pulsation of flow, the flow resistance will increase but at the same time, mass exchange between the mainstream and the boundary layer will also be enhanced, thereby enhancing heat transfer [30].

### 5.2. Performance at Different Cross-Sectional Areas

In the analysis of different flow channels, we can change the cross-sectional area of the flow channel by changing the width and height of the flow channel. It is expected that the uniformity of velocity distribution in the same passage will change when the cross-sectional area changes. The analysis results are shown in Figure 10. It can be seen from Figure 10a,b that the flow factor *f* decreases, and the variance of the velocity in the flow passage also decreases, indicating that the flow performance is better. It can be seen from Figure 10c that with the decrease of *j*, the heat transfer gradually decreases, indicating that the smaller the cross-sectional area, the better the heat transfer performance. Figure 10d

shows that the flow exchange performance of heat exchanger decreases with the increase of cross-sectional area.

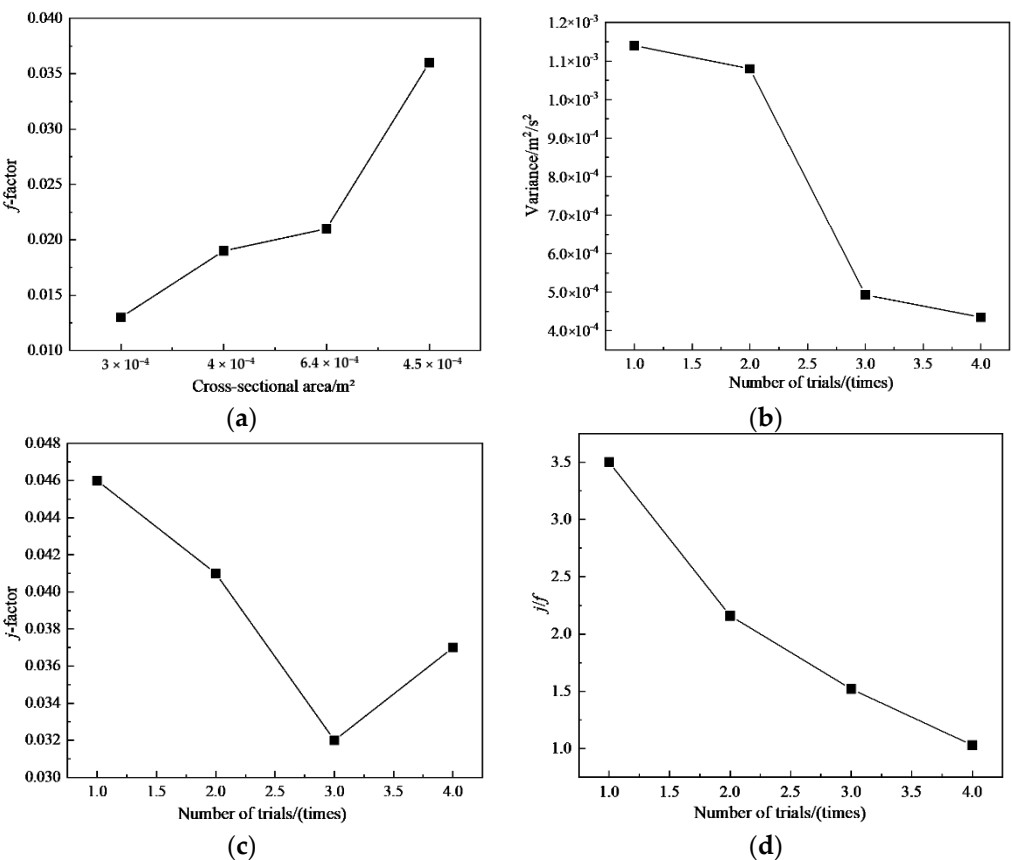

**Figure 10.** Study of the influence factor of oil cooler: (**a**) $f$-factor; (**b**) variance; (**c**) $j$-factor; (**d**) $j/f$.

### 5.3. Performance at Different Flow Path Lengths

It can be seen from Figure 11a,b that as the length of the flow passage increases, the velocity variance in the flow passage also decreases, and the fluid velocity distribution in the flow passage becomes more uniform. Therefore, in order to ensure the uniformity of flow rate in the flow channel, the length of the flow channel should be increased as long as possible on the premise of guaranteeing the heat transfer score. As can be seen from Figure 11a,c,d, the flow and heat transfer performance gets better and better with the increase of length.

### 5.4. Performance with Different Number of Flow Channel Layers

From Figure 12c,d, it is obvious that when the number of layer increases, the velocity of the fluid in the flow channel decreases obviously, and the flow rate in the same flow channel is more uniform, which is conducive to the heat transfer of the heat exchanger. In addition, the flow velocity distribution between different layers in these two working conditions needs to be studied and discussed further. Calculation results of flow field and temperature field of oil cooler are shown in Figure 12a,b. By comparing Figure 12c,d as well as the data can be found in Figure 13b, with the increase of flow channel layer, the average flow velocity of each flow layer not only changed, but also the speed of the fluid flow layer gap is in constant decreases.

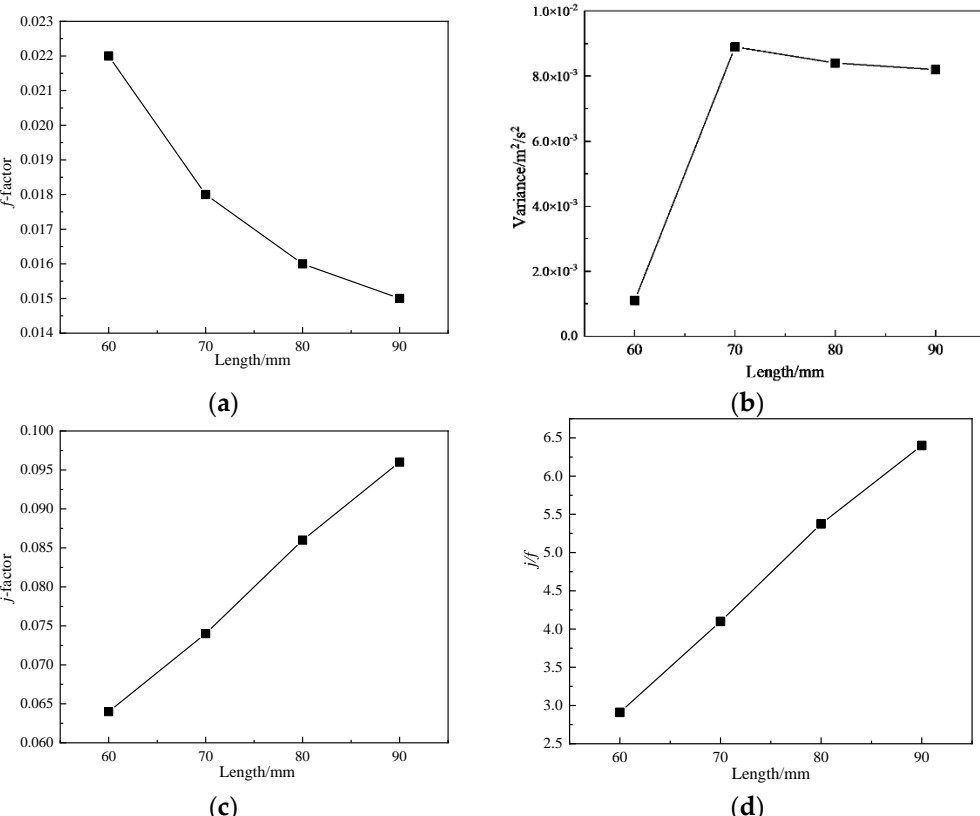

**Figure 11.** Study of the influence factor of oil cooler:(**a**) *f*-factor; (**b**) variance; (**c**) *j*-factor; (**d**) *j/f*.

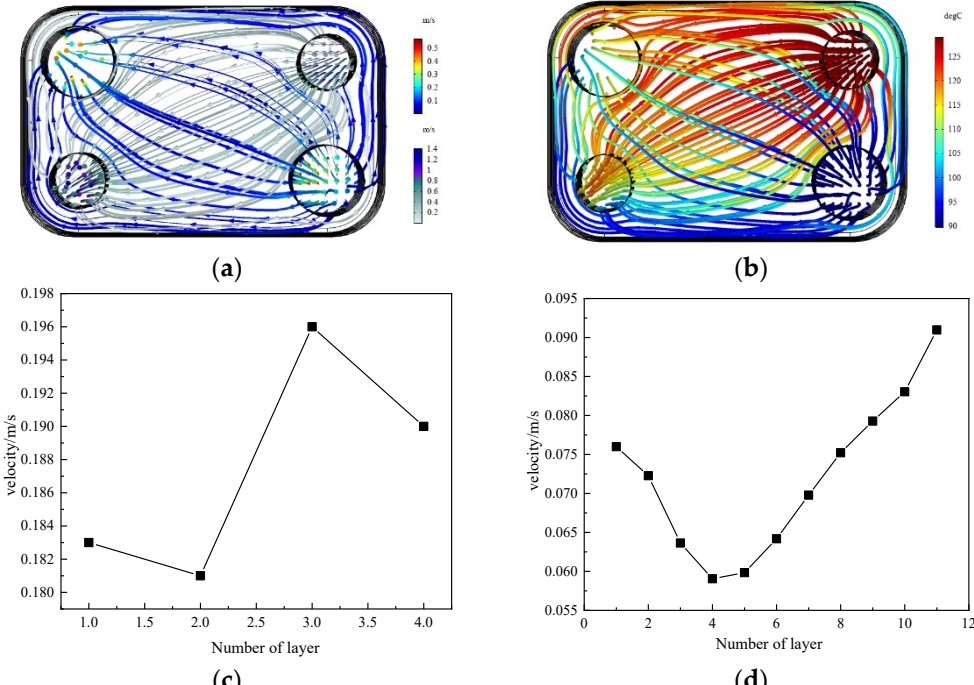

**Figure 12.** Velocity distribution diagram of oil cooler and flow channel velocity distribution diagram: (**a**) 4-layer heat exchanger speed distribution diagram; (**b**) 11-layer heat exchanger temperature distribution diagram; (**c**) Velocity distribution of flow channel with 4 layers; (**d**) Velocity distribution of flow channel with 11 layers.

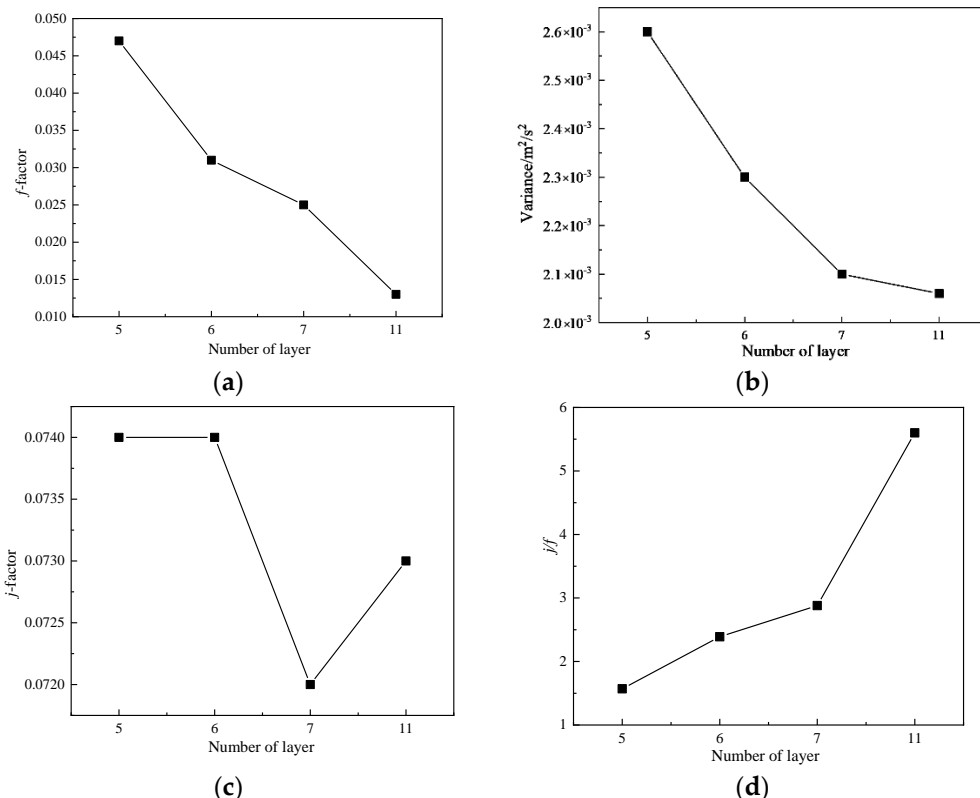

**Figure 13.** Study of the influence factor of oil cooler: (**a**) *f*-factor; (**b**) variance; (**c**) *j*-factor; (**d**) *j/f*.

It can be seen from Figure 13a,b that the number of flow channel layers has a certain relationship with the variance of the distribution of flow channel velocity. With the increase of the number of flow channel layers, the variance gradually decreases. Therefore, in the actual design process, under other conditions, the number of flow channel layers should be increased as many as possible to improve the flow performance. In addition, it can be seen from Figure 13c,d that with the decrease of *j*, the heat transfer gradually decreases, indicating that the smaller the cross-sectional area, the better the flow and heat transfer performance.

### 5.5. Optimization of Structural Parameters

Through the ratio of *j/f* index of each part under different structural parameters, it can be concluded that when the cross-sectional area is 3 mm², it is 142% higher than the original parameter, when the length is 90 mm, it is 119% higher than the minimum parameter, and when the number of layers is 11, it is 134% higher than the original parameter. Therefore, the optimal structural parameters are proposed as follows: cross-sectional area is $3 \times 10^{-4}$ mm², length is 90 mm, number of layers is 11. According to the calculation in Figure 14a,b, under this parameter, the heat transfer is increased by 47%, and with the total pressure drop increased by only 30%.

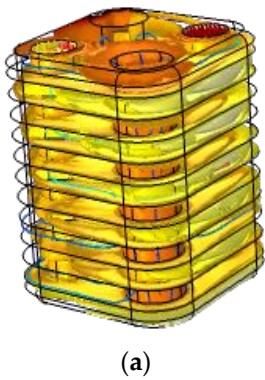 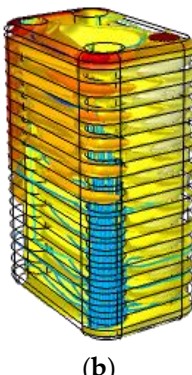

(**a**)　　　　　　　　　　　　　　　(**b**)

**Figure 14.** Comparison of heat exchange between basic and improved parameters: (**a**) Heat exchange after basic parameter setting; (**b**) Heat exchange after improved parameter setting.

## 6. Conclusions

In this paper, the equivalent simulation method of the automotive oil cooler is studied based on the non-uniform permeable flow model with a multi-scale approach. Firstly, the unit heat transfer model in different directions is established and calculated, the anisotropic flow fitting correlation is obtained. Under this base, the overall equivalent model is simplified with non-uniform permeable flow model and local thermal non-equilibrium model. A few interesting conclusions can be obtained, as follows:

(1) First, a multi-scale coupling method based on unit heat transfer model is proposed to simulate the flow and heat transfer performance of heat exchanger. The flow of the whole heat exchanger is simulated by the non-uniform seepage flow model, and the heat transfer is simulated by the local thermal non-equilibrium model.

(2) Next, a vehicular oil cooler is used to verify the effectiveness of this method. By comparing with the experimental results, the maximum error of this equivalent simulation model for flow and heat transfer under different working conditions is 9.2%, which proves the validity of the equivalent model.

(3) Finally, the flow and heat transfer performance under different structural parameters was studied. At the same time, the best structural parameters could applicable to the present oil cooler are proposed, namely: cross-sectional area of $3 \times 10^{-4}$ mm$^2$, length of 90 mm, number of layers is 11. Comparing with the original structure, the heat transfer performance is increased by 47%, while the total pressure drop increased by only 30%.

**Author Contributions:** Conceptualization, J.F.; Methodology, Z.H. and Y.Z.; Resources, G.L.; Software, Y.Z.; Supervision, J.F.; Writing—original draft, Z.H.; Writing—review & editing, J.F. All authors have read and agreed to the published version of the manuscript.

**Funding:** This research was supported by Zhejiang Provincial Natural Science Foundation of China under Grant No. LQ20E060003; Scientific Research Foundation of Zhejiang University City College (No. J-202116, X-202205); Projects of Hangzhou Agricultural and Social Development Research (No. 20201203B128, 20212013B04) and the 2021 Teacher Professional Development Program for Domestic Visiting Scholars in universities (No. FX2021105).

**Institutional Review Board Statement:** Not applicable.

**Informed Consent Statement:** Not applicable.

**Data Availability Statement:** Not applicable.

**Conflicts of Interest:** The authors declare no conflict of interest.

## Abbreviations

| | |
|---|---|
| $\mu$ | Dynamic viscosity |
| **u** | Velocity vector |
| $\rho$ | Fluid density |
| $p$ | Pressure |
| **I** | Identity orthogonal matrix |
| $\varepsilon_p$ | Void fraction |
| $\kappa$ | Permeability of porous media |
| $C_p$ | Specific heat capacity |
| $T_{in}$ | Inlet temperature |
| $Q_m$ | Quality of the source |
| **F** | Volume force |
| **κ** | Porosity matrix |
| $C_F$ | Dimensionless Faux-Hemmel |
| $\theta_s$ | Volume fraction of a solid |
| $k_s$ | Thermal conductivity of solids |
| $k_f$ | Thermal conductivity of a liquid |
| $T_{out}$ | Outlet temperature |
| $T$ | Temperature |
| $Q$ | Heat exchange amount, total heat exchange of heat exchanger |
| $\rho_s$ | Solid density |
| $\rho_f$ | Liquid density |
| $m$ | Mass quality |
| $K$ | Heat transfer coefficient of fluid |
| $A$ | Heat exchange area, cross-sectional area |
| $\Delta T$ | Temperature difference between the inlet and outlet of the hot side of the oil cooler |
| $Nu$ | Nusselt number |
| Re | Reynolds number |
| Pr | Prandtl number |
| $v_m$ | Average velocity of the fluid in the flow channel |
| $L$ | Length of flow channel |
| $\Delta p$ | Pressure difference on both sides of the flow passage |

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
