# Peer review of "Numerical Study and Structural Optimization of Vehicular Oil Cooler Based on 3D Impermeable Flow Model"

_sustainability, doi:10.3390/su14137757_

Round 1

Reviewer 1 Report

1. In the Introduction section, it is necessary to present more recent and new articles related to the study. Please improve the Introduction section.

2. Include references to mathematical expressions that are not derived from this work.

3. Figure quality is poor. Please improve the figure quality.

4. The explanations and discussions presented for Figures are inadequate. Provide a better understanding of the results by adding more detailed discussion and explanation.

5. The conclusion section is weak and inadequate. Please rewrite it in its entirety.

6. Correct the typos in the entire text and improve the English level of the article.

7. Organize references in accordance with journal standards.

Author Response

With thanks, we appreciate the valuable comments/suggestions given by the Reviewers. These comments/suggestions have stimulated us to introduce revisions and thus improve the quality of our paper. We have taken into account all comments and made the revisions, which are highlighted with “Track Changes” function. The comments quoted by the reviewers are responsed with detail in attachment.

Reviewer 2 Report

  The cooling systems of vehicles and equipment are extensive and contain many components, for example a truck can have several radiators.
The technical parameters of coolers often depend on the environmental conditions in which the vehicle is operating.
There are many articles in the literature in which cooling systems are modeled for optimization.
It is obvious that the heat given off by the cooler can be treated as losses in terms of efficiency.
I propose to describe a typical cooling system, write a mathematical model, write detailed models for individual elements (e.g. water coolers, oil coolers, retarder coolers, etc.).
Then show the model, calculation results and measurement results on a real system.
Your article is very interesting and I appreciate your hard work.
I believe that you only need to show the context of your calculations.
Good luck!
For example, see the article: DOI 10.17531 / ein.2019.4.6, you will find links to similar articles in the reference list in this article.

Author Response

(The authors gave the same response as above.)

Reviewer 3 Report

The paper describes a methodology to model the behavior of heat exchangers, which is interesting from a design perspective. Then the model is used to perform a parametric sensitivity and identify trends for possible optimization. The modelling is interesting but the paper is poorly organized and full of typos. In addition, there are many charts but it's hard to get an idea of the impact on the solutions. More contours should be provided to explain the results.

Specific comments:

Abstract: "By comparison between different models, and ultimately to select the optimal model." -> this sentence makes no sense

93: , -> .

120: space after comma, space after dot

142: space between structure and (left)

155: weird font for the first "a" in "analysis"

160: Figure.3 -> Figure 3

284: The Nusselt -> the Nusselt

292-293: improper use of ;

304-308: too long sentence. What means "USES"?

328: 1ab -> 11b

363: missing number before mm2

367: missing units for area

368: what about the pressure drop?

402: p is subscript

Many of these errors are repeated and not reported here above.

Author Response

(The authors gave the same response as above.)

Round 2

Reviewer 1 Report

The article can be accepted as it is.

Author Response

Dear Reviewer:

  Thank you very much for your affirmation and advices. We have checked all the punctuation, typos,  grammar, spaces and points and revised them carefully again. For details, please see the manuscript.

Best regards

Reviewer 3 Report

The paper is still full of typos, grammar issues, multiple spaces, missing or repeated points. Despite that, the work is interesting and if these aspects are fixed, it can be published.

Author Response

(The authors gave the same response as above.)
